# CDC42-IQGAP Interactions Scrutinized: New Insights into the Binding Properties of the GAP-Related Domain

**DOI:** 10.3390/ijms23168842

**Published:** 2022-08-09

**Authors:** Niloufar Mosaddeghzadeh, Silke Pudewell, Farhad Bazgir, Neda S. Kazemein Jasemi, Oliver H. F. Krumbach, Lothar Gremer, Dieter Willbold, Radovan Dvorsky, Mohammad R. Ahmadian

**Affiliations:** 1Institute of Biochemistry and Molecular Biology II, Medical Faculty and University Hospital Düsseldorf, Heinrich Heine University Düsseldorf, 40225 Düsseldorf, Germany; 2Institute of Physical Biology, Heinrich Heine University Düsseldorf, 40225 Düsseldorf, Germany; 3Institute of Biological Information Processing, Structural Biochemistry (IBI-7), Forschungszentrum Jülich, 52425 Jülich, Germany; 4Center for Interdisciplinary Biosciences, P. J. Šafárik University, Jesenná 5, 040 01 Košice, Slovakia

**Keywords:** CDC42, GAP, GAP-related domain, GRD, GTPase activating protein, IQGAP, nucleotide-independent binding, RASGAP, RHO GTPases, scaffold protein, scaffolding protein, switch regions

## Abstract

The IQ motif-containing GTPase-activating protein (IQGAP) family composes of three highly-related and evolutionarily conserved paralogs (IQGAP1, IQGAP2 and IQGAP3), which fine tune as scaffolding proteins numerous fundamental cellular processes. IQGAP1 is described as an effector of CDC42, although its effector function yet re-mains unclear. Biophysical, biochemical and molecular dynamic simulation studies have proposed that IQGAP RASGAP-related domains (GRDs) bind to the switch regions and the insert helix of CDC42 in a GTP-dependent manner. Our kinetic and equilibrium studies have shown that IQGAP1 GRD binds, in contrast to its C-terminal 794 amino acids (called C794), CDC42 in a nucleotide-independent manner indicating a binding outside the switch regions. To resolve this discrepancy and move beyond the one-sided view of GRD, we carried out affinity measurements and a systematic mutational analysis of the interfacing residues between GRD and CDC42 based on the crystal structure of the IQGAP2 GRD-CDC42^Q61L^ GTP complex. We determined a 100-fold lower affinity of the GRD1 of IQGAP1 and of GRD2 of IQGAP2 for CDC42 mGppNHp in comparison to C794/C795 proteins. Moreover, partial and major mutation of CDC42 switch regions substantially affected C794/C795 binding but only a little GRD1 and remarkably not at all the GRD2 binding. However, we clearly showed that GRD2 contributes to the overall affinity of C795 by using a 11 amino acid mutated GRD variant. Furthermore, the GRD1 binding to the CDC42 was abolished using specific point mutations within the insert helix of CDC42 clearly supporting the notion that CDC42 binding site(s) of IQGAP GRD lies outside the switch regions among others in the insert helix. Collectively, this study provides further evidence for a mechanistic framework model that is based on a multi-step binding process, in which IQGAP GRD might act as a ‘scaffolding domain’ by binding CDC42 irrespective of its nucleotide-bound forms, followed by other IQGAP domains downstream of GRD that act as an effector domain and is in charge for a GTP-dependent interaction with CDC42.

## 1. Introduction

RHO GTPases act, with some exceptions [1], as molecular switches by cycling between an inactive (GDP-bound) and an active (GTP-bound) state. Their functions at the plasma membrane are usually controlled by three groups of regulatory proteins: guanine nucleotide dissociation inhibitors (GDIs), guanine nucleotide exchange factors (GEFs) and GTPases activating proteins (GAPs) [2]. The formation of the active GTP-bound state of RHO GTPases, such as CDC42, is accompanied by a conformational change in two regions, known as switch I and II (encompassing amino acids or aa 29–42 and 62–68, respectively); these regions provide a platform for a GTP-dependent, high-affinity association of structurally and functionally diverse effector proteins, e.g., ACK, PAK1, WASP, ROCKI, DIA and IQGAP1, through their so-called GTPase-binding domains (GBDs) [3,4,5,6,7,8,9,10,11,12,13] (reviewed in [14]). GTPase-effector signaling activates further a wide variety of pathways in all eukaryotic cells [2].

A unique feature distinguishing the RHO family from other small GTPase families is the presence of a 12 amino-acid insertion (aa 124–135 in CDC42) that protrudes from the G domain structure by forming a short helix, the so-called insert helix (IH) [15]. This IH is highly charged and variable among the members of the RHO family [15]. The IH has been very recently shown to have larger conformational flexibility in the GDP-bound CDC42 than in the GTP-bound CDC42 [16]. IH is a binding site for RHOGDI1, p50GAP, DIA, FMNL2, PLD1 and IQGAP2 [10,12,17,18,19,20,21], and appears rather essential for downstream activation of RHO GTPases [21,22,23].

IQGAP1 is ubiquitously expressed and shares a similar domain structure with its human paralogs IQGAP2 and IQGAP3 (Figure 1A), including an N-terminal calponin homology domain (CHD), a coiled-coil repeat region (CC), a tryptophan-containing proline-rich motif-binding region (WW), four isoleucine/glutamine-containing motifs (IQ), a RASGAP-related domain (GRD), a RASGAP C-terminal domain (RGCT) and a very C-terminal domain (CT). IQGAPs interact with a large number of proteins and modulate the spatiotemporal distributions of distinct signal-transducing protein complexes [24,25,26,27,28,29,30,31,32,33,34]. As multidomain scaffold proteins, they safeguard the magnitude, efficiency and specificity of signal transduction [35]. They have been localized at multiple subcellular sites orchestrating different signaling pathways and thus controlling a variety of cellular functions [36,37,38,39,40,41,42]. Notably, IQGAP1 has been implicated as a drug target due to its vital regulatory roles in cancer development [42,43,44,45,46,47,48,49] although the molecular mechanism of its functions is unclear.

Earlier studies analyzed the crucial role of IQGAP RGCT in high-affinity binding to the switch regions of the GTP-bound, active CDC42 and proposed it as an IQGAP ‘effector domain’ [5,50,51]. Accordingly, Swart-Mataraza et al. reported that the CDC42 GppNHp can still bind to IQGAP-ΔGRD (lacking aa 1122–1324) [52]. Moreover, Li et al. mapped the CDC42 and IQGAP binding regions and determined that switch I and surrounding regions (residues 29–55) together with the insert region (residues 122–134) are required for high affinity binding to IQGAP1 [53]. LeCour et al., however, solved a crystal structure of constitutively active CDC42(Q61L) in complex with the IQGAP2 GRD (GRD2) and proposed that CDC42 binds GRD2 from two different sites in a 4:2 stoichiometry [12,54]. One is the ‘GAPex-mode binding site’ (ex stands for ‘extra’ subdomains consisting of variable N- and C- terminal flanking regions) and the other is the ‘RASGAP-mode binding site’ very much resembling the RASGAP and CDC42GAP structures [18,55] with a conserved core domain (GAPc). Analyzing this structure, Ozdemir et al. proposed that CDC42 IH binding to the GAPex-domain induces GRD2 dimerization and changes the RASGAP site allosterically, which subsequently create another interaction interface for CDC42 binding (leading to a 2:1 stoichiometry of GRD2 and CDC42) [54].

A number of biophysical and biochemical studies have provided valuable insights into the structural and binding properties of the C-terminal domains of IQGAP1 (C794) and IQGAP2 (C795), encompassing the GRD, RGCT and CT domains, with CDC42 [12,50,52,53,54,56,57,58,59,60,61,62,63]. Evidently, all three domains bind with different affinities to CDC42 [5]. However, the mechanistic principles behind these interactions have remained unclear. Moreover, there are conflicting views regarding the assignment of a ‘CDC42-specific GBD’ for IQGAPs. One model proposes the GRD and its RASGAP-mode binding with the switch regions of CDC42 [12,54,56,58,64], whereas the other model excludes GRD and marks RGCT, located distal to the GRD, as crucial for high-affinity binding to CDC42 in a GTP-dependent manner [5,26,50,51,52]. Aiming to shed light on this discrepancy and to understand the molecular basis of CDC42-IQGAP interaction we comprehensively investigated the nature of the GRD interaction with CDC42 in this study and determined the role of the IH of CDC42 in contributing to GRD association. Furthermore, we studied the binding characteristics of C794 regarding the switch region and IH contact sites by mutational analysis, and verified the results in cell-based studies with endogenous IQGAP1. Collectively, our results consolidate and refine the importance of IQGAP RGCT as the true GBD in the recognition of CDC42 and its binding in a GTP-dependent manner. The GRD, although not a central effector domain, is evidently necessary for scaffolding CDC42 and facilitating its recruitment to preexisting cues.

## 2. Results and Discussion

IQGAP1 and IQGAP2 proteins were analyzed in this study to critically evaluate the function of the respective GRD domains. First, we determined the CDC42 binding properties of different IQGAP proteins, including IQGAP1 full-length (FL). Second, we examined the role of amino-acids crucial for the interplay between IQGAP2 and CDC42 using mutational IQGAPs and CDC42 variants. Third, we analyzed the impact of CDC42 IH as an IQGAP binding site. Fourth, we investigated the RASGAP activity of IQGAP1 GRD towards eight different members of the RAS family and examined the introduction of a catalytic arginine finger in the GRD.

### 2.1. GRD Is Not the Prominent Binding Domain for High IQGAP-CDC42 Affinity

#### 2.1.1. GRD Binds to CDC42 with Very Low Affinity in a Nucleotide-Independent Manner

Different domains and fragments of the IQGAPs, including GRD1 and C794 of IQGAP1, as well as GRD2 and C795 of IQGAP2 (Figure 1A), were purified to determine their binding affinities for mGDP- and mGppNHp-bound CDC42 using fluorescence polarization. Obtained dissociation constants (K_d_; Figure 1B) clearly show that all IQGAP constructs are able to bind CDC42 but with different affinities and preferences for the nucleotide-bound forms of CDC42. GRDs of both IQGAPs are low-affinity binders and do not discriminate between the active and the inactive states of CDC42. Similar observations were made for GRD3 and the CT of IQGAP1 (Appendix A). In contrast, C794 and C795, encompassing in addition to both GRD and CT also the central RGCT (Figure 1A), exhibited K_d_ values of 0.6 and 0.9 µM, respectively, indicating an around 100-fold higher affinity for the GTP-bound active CDC42 as compared to CDC42 GDP (Figure 1B). This result clearly suggests that RGCT but not GRD represents a ‘CDC42-specific GBD’ for at least IQGAP1 and IQGAP2, by directly associating with the switch regions of CDC42 GTP. Unfortunately, our efforts to obtain IQGAP1 RGCT (aa 1276–1575) and IQGAP3 C790 (aa 841–1631) for determining their binding affinities to the members of the RHO GTPase family, including CDC42, has been remaining unsuccessful [26,51]. Purified IQGAP1 RGCT tends to assemble into higher oligomeric or polymeric states, and, thus, is disabled in binding CDC42 [51].

Several lines of evidence support the crucial role of RGCT rather than GRD as the IQGAP effector domain for CDC42: (i) Here we can show that proteins containing RGCT bind with a more than 100-fold affinity to CDC42 mGppNHp as compared to isolated GRD or CT (Figure 1B and Appendix A), (ii) substitution of the Serine 1443 for glutamate (a phosphomimetic mutation) drastically impaired IQGAP1 binding to CDC42 mGppNHp [5,51]; (iii) an IQGAP1 protein, lacking the GRD (aa1122–1324), only binds CDC42 GppNHp, in contrast to IQGAP1 itself, that binds both GppNHp-bound and GDP-bound CDC42 [52]. The latter has been also demonstrated in other studies [63,64] and support our previous [26,51] and current findings that IQGAP domains, including GRD and CT, bind CDC42 GDP as strong as CDC42 GppNHp (Figure 1B).

#### 2.1.2. Endogenous IQGAP1 also Binds CDC42 GDP

Serum-stimulated HEK293 cells, endogenously expressing IQGAP1 full-length (FL), were now used to carry out a pull-down assay with purified GST-fusion proteins of CDC42 and RAC1 in either GDP-bound or GppNHp-bound forms. IQGAP1 FL bound to these GTPases, regardless of their nucleotide status even though the binding to GDP-bound proteins was observed to be much weaker than the GppNHp-bound proteins (Figure 1C). This pattern corresponds to the binding behavior of C794 and not with the binding of GRD1 alone. Densitometric evaluation of three independent pull-down experiments showed that IQGAP1 FL binding to CDC42 GDP is much stronger than to RAC1 GDP (Figure 1C).

Altogether, our data suggest that IQGAP1 forms a complex with CDC42 through different sites in both nucleotide-dependent and nucleotide-independent manner.

### 2.2. Switch Regions of CDC42 Are Not the Main Binding Sites for the GRDs

Timpson’s and our group have provided evidence that the IQGAP RGCT is essential for high affinity binding to the switch regions of the GTP-bound, active CDC42 and thus acts as an IQGAP ‘effector domain’ [5,50,51]. This critical issue has now been further expanded with additional experiments as described above (Figure 1), and confirms the crucial role of the RGCT as an IQGAP ‘effector domain’ that selectively associates with CDC42 GTP and carries out the high affinity association. Other groups have, in contrast, used the constitutive active CDC42(Q61L) in their structural and biochemical analysis and proposed that CDC42(Q61L) GTP GRD forms a GTPase-effector complex [12,54,56,57]. Such a role of the GRD in associating with CDC42 GTP is astonishing considering the afore mentioned studies on both GRD1-CT that binds CDC42 with a higher affinity as compared with GRD and an IQGAP1 variant, lacking the RASGAP domain (aa 1122–1324), which equally interacts with CDC42 as compared with IQGAP1 wild type [52]. To clarify this discrepancy, we have carefully examined ‘the RASGAP-mode binding site’ of CDC42 using mutational approaches coupled with kinetic and equilibrium measurements. Results of this examination are discussed in following subsections.

#### 2.2.1. Mutations in CDC42 Switch Regions Only Mildly Affect GRD Binding

Proposed interacting mode of GRD with the switch regions of CDC42 (RASGAP mode binding) was deduced from the IQGAP2 GRD2 structure in complex CDC42^Q61L^ GTP [12] and two CDC42 mutation variants within the switch I and II regions (2xSW and 8xSW) and a 11-residues mutant variant within the GRD of IQGAP2 C795 (11xGRD) were generated as illustrated in Figure 2A. Identical and highly conserved residues within the interacting interface highlighted in Figure 2B, were all replaced by alanine. All variants were stable in their purified forms and Far-UV CD spectroscopic measurements excluded any improper folding as compared to the wild-type proteins (Appendix A).

We first determined the K_d_ values for the GRD1 and GRD2 interaction with the mGppNHp-bound CDC42 WT, 2xSW and 8xSW. Interestingly, we found a two to three-fold reduction in the binding affinity of GRD1 but no notable reduction for GRD2 with the CDC42 variants as compared to CDC42 WT (Figure 1B, Figure 2C and Appendix A). As the effect of 2x and 8x introduced mutations on the proposed crucial interaction sites of CDC42 and GRD2 did not result in a decrease of affinity, our data clearly indicates that the association of CDC42 switch regions with IQGAP must be through other sites rather than the GRD.

#### 2.2.2. IQGAP C794/C795 Binding Is Impaired by Switch Region and GRD Mutations

Next, we measured the K_d_ values for the interaction of IQGAP1 C794 or IQGAP2 C795, containing the GRD, RGCT and CT domains, with mGDP-bound and mGppNHp-bound CDC42 variants. Data shown in Figure 2D (Appendix A) indicate that the substitution of two amino acids in the switch regions was not sufficient to largely impair the CDC42-C794 interaction. However, mGppNHp-bound, but not mGDP-bound CDC42 8xSW exhibited a drastic reduction (86-fold) in its binding affinity for C794. For mGDP-bound CDC42, introduction of SW mutations only slightly decreased the affinity of C794. The IQGAP2 C795 binding to the CDC42 switch regions was not impaired by neither 2x nor 8x mutants of CDC42 in mGppNHp-bound state. Interestingly, IQGAP2 C795 showed a slightly decreased binding to the mGDP-bound CDC42 2xSW mutant but no binding to the 8xSW mutant, a much different result than obtained for GRD2 binding alone. The data from real-time stopped-flow fluorescence spectrometry (Figure 2E and Appendix A) showed both IQGAPs associated with similar k_obs_ values, as observed in Figure 2D.

The next question addressed was to what extent CDC42 binding of IQGAP1 FL was affected by the switch region mutations. Therefore, endogenous IQGAP1 was pulled down from HEK293 lysates using GDP-bound and GppNHp-bound GST-CDC42 WT, 2xSW and 8xSW. As shown in Figure 2F, IQGAP1 binding to CDC42 did not change with two amino acid substitution of the switch regions but was disrupted with the eight mutations. These experiments support our kinetic and equilibrium measurements and clearly indicate that the switch regions are significant for the IQGAP1 interaction with both GDP-bound and GppNHp-bound CDC42.

Taken together, the presented data suggest a slightly different binding behavior of IQGAP1 and IQGAP2 variants for CDC42. Our results do not support the interacting mode between IQGAP and CDC42 based on the crystal structure [12] and the central role of the GRD in it [54] since the introduction of SW mutations of CDC42 clearly affected C795/C794 binding but only little the GRD binding. We, in contrary, propose that the interactions sites on IQGAP for complex formation with CDC42 GTP are clearly within the RGCT and might be different between IQGAP1 and IQGAP2.

### 2.3. Insert Helix Contributes to the Binding Affinity of CDC42 for IQGAP1 GRD

The question arises as which regions on CDC42 could bind GRD if we can now exclude the switch regions. A region/site that has attracted our attention is the IH of CDC42 for valid reasons. We have shown that IQGAPs bind to RAC-like and CDC42-like proteins but not to the other members of the RHO family [26] and the IH consistently is a highly variable region among the RHO GTPases (Figure 3A) [15]. Several CDC42-binding proteins, including RHOGDI1, p50GAP, FMNL2 and IQGAP2 have been shown to contact the IH [10,12,17,18,20]. Thus, mutational analysis of the CDC42 IH was performed, using four different single residue mutations and a quadruple mutation (Figure 3A and Appendix A). Note that variable residues were replaced in CDC42 by the corresponding residues of RAC1. Most remarkably and in sharp contrast to the SW mutations (Figure 2), all IH mutations abolished GRD1-CDC42 interaction irrespective of the nucleotide-bound states of CDC42 (Figure 3B and Appendix A), which underlines the central role of CDC42 IH in GRD binding. The scenario was rather different for C794, which binds mGDP-bound CDC42 with 3-fold and mGppNHp-bound CDC42 with 20-fold lower affinities (Figure 3B). These data are consistent with the recent observation by Nussinov and colleagues that the CDC42 IH reveals nucleotide-dependent conformational flexibility [16].

The data from fluorescence polarization could be verified via pull-down assay. The binding pattern of CDC42 IH mutants with endogenous IQGAP1 followed the same pattern, displaying no binding for A130K and 4xIH and very weak binding for S124D (Figure 3C). Generally, binding could be observed much stronger for GppNHp-bound than for GDP-bound CDC42 variants, supporting the pull-down data shown above (Figure 1C).

Several published studies have shown that mutations of the CDC42 IH impact their properties in binding IQGAPs. Li et al. (1999) have shown that IH deletion in CDC42 impairs its binding affinity for the effectors, in particular IQGAP1 C794 [53]. Owen et al. (2008) investigated the impact of the IH mutations in CDC42^Q61L^ on IQGAP1 C794 binding [56]. Consistent with our findings, they observed a slight decrease in C794 affinity for CDC42^Q61L^ with A130K or N132K. Moreover, Ozdemir et al. also applied the CDC42^Q61L^ variant and suggested the IH together with switch I region to be mainly responsible for its binding to the *ex*-domain of GRD (GRDex) of IQGAP2 [54].

### 2.4. Q61L Variant Is Not a Wildtype Equivalent for CDC42-IQGAP Interactions

An issue that still needs to be addressed is why are there several discrepancies between the studies regarding the GRD binding property for CDC42? A possible answer to this question is the use of different CDC42 mutants in these studies that are alike, but not equivalent, especially regarding this interaction.

In the GTP-bound CDC42, Q61 acts as a ‘catalytic residue’ that is involved in hydrogen bonding with a catalytic water molecule, an arginine finger of GAP and the γ-phosphate of GTP, initiating a nucleophilic attack that hydrolyzes GTP (Figure 4A) [18,65]. L61 does not, however, undergo these functionally critical hydrogen bonds but rather points towards protein surface without causing significant structural changes (Figure 4A). As a result, the substitution of Q61 by leucine drastically increases the binding affinity of IQGAPs for CDC42^Q61L^ GTP by up to 15-fold as was clearly demonstrated previously [5,26,51]. Despite this fact, many groups use this CDC42 variant for the interaction analysis of effectors, such as IQGAPs [12,54,56,57]. Thus, we revisited this issue and have comparatively analyzed the interaction of IQGAP1 GRD with CDC42^Q61L^ and CDC42^wt^ using fluorescence polarization and size exclusion chromatography (SEC). Equilibrium measurements shown in Figure 4B clearly revealed that the Q61L mutation results in a strong enhancement of GRD1 and GRD2 binding with the mGppNHp-bound CDC42, but not with mGDP-bound CDC42. The binding affinity of mGppNHp-bound CDC42^Q61L^ rises from a low affinity 186 µM/69 µM binding to a high 2.7 µM/2.5 µM binding for GRD1/GRD2, respectively (Figure 4B and Appendix A). This is a change of 30–50-fold and might explain the huge differences of CDC42 interactions with GRD. Moreover, SEC analysis showed that GRD1 forms a 2:2 stoichiometry with CDC42^wt^ GppNHp but 2:1 stoichiometry with CDC42^Q61L^ GppNHp (Figure 4C–F).The latter is remarkably consistent with the previous reports on a high-affinity binding of IQGAP2 GRD2 with CDC42^Q61L^ GTP and 4:2 and 2:1 stoichiometry, respectively [12,54]. These findings verified the clear difference between CDC42^wt^ and CDC42^Q61L^ and how replacement of Q61 by L changes the binding properties (affinity and stoichiometry) of CDC42 interaction with IQGAP GRDs.

Chen et al. have reported that the Q61L mutation strengthen hydrogen bond interactions between CDC42 and the γ-phosphate of GTP [66]. Analyzing the Cdc42^Q61L^ GTP GRD2 structure, Ozdemir et al. proposed that CDC42 IH binding to the GAPex-domain induces allosteric changes in the RASGAP site, which in turn facilitate GRD dimerization, and enable the second CDC42^Q61L^ to bind to this site (yielding a 2:1 stoichiometry) [54]. Collectively, we recapitulate that CDC42^Q61L^ is not an ideal analog of CDC42^wt^ especially in studying the interaction of the downstream effectors. G12V and Q61L mutations of CDC42 cause GAP insensitivity leading to sustained hyperactivation of CDC42 [16,18,55,65,66]. Thus, we suggest CDC42^wt^ GppNHp and even CDC42^G12V^ GTP variants as more suitable species for the investigation of CDC42-effector interaction rather than CDC42^Q61L^ GTP.

### 2.5. GRD Lacks the Structural Fingerprints to Induce the GAP Activity

The structure of the RAS-RASGAP complex shows GAP-334 interacting predominantly with the switch regions of RAS [55]. Three regions (finger loop, FLR motif and helix α7/variable loop) constitute structural fingerprints of the RASGAP p120 and neurofibromin that form critical RAS binding sites in order to apply an arginine finger into the active center of RAS [67,68]. Amino acid sequence analysis of these RASGAPs with the three IQGAP paralogs showed that major parts of these fingerprints are different in IQGAPs (Figure 5A). Moreover, the catalytic arginine is missing and there is instead a threonine (T1045 in IQGAP1; Figure 5A). Thus, it is quite understandable why IQGAP1 did not display RASGAP activities towards HRAS [60]. It is, however, known that GAPs specific for other members of the RAS superfamily use other catalytic residues than an arginine (reviewed in [69,70]).

We set out to examine a possible GAP activity of IQGAP1 GRD towards different RAS family GTPases. Figure 5B shows that IQGAP1 GRD is a pseudo-RASGAP domain with no obvious catalytic ability (orange bars). Earlier studies have shown that the substitution of the arginine finger of the RASGAPs to other amino acids completely abolishes their GAP activity [67,68]. Therefore, threonine 1046 of IQGAP1 GRD was replaced by an arginine and the impact of T1046R on the GTP hydrolysis of the eight RAS proteins was measured. Data shown in Figure 5B revealed no apparent GAP activities of IQGAP1 GRD^T1046R^ (green bars) as expected for a RASGAP. These data suggest that IQGAPs, besides lacking an arginine finger, do not contain critical RAS-binding residues of the a7/variable loop (Figure 5A).

## 3. Material and Methods

### 3.1. Constructs 

The pGEX4T1 encoding an N-terminal glutathione S-transferase (GST) fusion protein was used to overexpress human IQGAP1 (accession number P46940) GRD1 (aa 962–1345), C794 (aa 863–1657) and CT (aa 1576–1657); human IQGAP2 (accession number Q13576) GRD2 (aa 875–1246) and C795 (aa 780–1575); human IQGAP3 (accession number P60953) GRD3 (aa 942–1330); human CDC42 (accession number P60953; aa 1–178). All constructs and related variants are list in Appendix A. For purification of these proteins, pGEX-4T1 constructs were transformed in *Escherichia coli* and proteins were isolated via affinity chromatography using a glutathione Sepharose column on a ÄKTA start protein purification system (Cytiva, US) [71]. GST-cleavage was carried out by incubation with thrombin (#T6884-1KU, Sigma Aldrich, Taufkirchen, Germany) at 4 °C until full digestion of the fusion protein. Quality of the proteins were checked via SDS-PAGE and Coomassie staining. CDC42 variants were further verified for their activity in HPLC by determining the amount of bound nucleotide [71]. Nucleotide free proteins were prepared by incubating the proteins with alkaline phosphatase (#P0762-250UN, Sigma Aldrich, Germany) and phosphodiesterase (#P3243-1VL, Sigma Aldrich, Taufkirchen, Germany) at 4 °C [71]. CDC42 variants were labelled with either GDP (#51060, Sigma Aldrich, Taufkirchen, Germany), GppNHp (#NU-401, Jena Bioscience, Jena, Germany), mant-GDP (#NU-204, Jena Bioscience, Jena, Germany) or mant-GppNHp (#NU-207, Jena Bioscience, Jena, Germany).

### 3.2. Circular Dichroism (CD) Spectrometry 

Far-UV-CD spectroscopy of protein samples were performed on a JASCO J-715 CD spectropolarimeter (Jasco, Gross-Umstadt, Germany) using quartz cuvettes (Helma, Mühlheim, Germany) with 1 mm path length. Spectra were recorded at protein concentrations of 20 µM CDC42 WT and variants in 1 mM NaPi buffer, pH 7.0 or 8 µM IQGAP WT and variants in 12.5 mM TRIS/HCl pH 7.4, 37.5 mM NaCl, 1.25 mM MgCl_2_, at 22 °C with instrument settings as follows: 0.1 nm step size, 50 nm min^−1^ scan speed, 1 nm band with. Signal-to-noise ratio was improved by accumulation of 10 scans per sample. The mean residue ellipticity ${\left[\theta \right]}_{\mathrm{mrw}}$ in deg·cm^2^·dmol^−1^ was calculated from the equation ${\left[\theta \right]}_{\mathrm{mrw}}=\left(\theta_{\mathrm{obs}}\times \mathrm{MRW}\right)/\left(c\times d\times 10\right)$, with $\theta_{\mathrm{obs}}$, observed ellipticity (in degrees); c, concentration (in g/mL); d, cell path length (in cm); MRW (mean residue weight), molecular weight divided by number of peptide bonds.

### 3.3. Cell Culture and Lysis 

HEK293 cells were cultured in Dulbecco′s Modified Eagle′s Medium (DMEM) (# 12320032, Thermo Fisher, Waltham, CA, USA) supplemented with 10 % FBS and 1% Penicillin/Streptomycin in an exponential growth phase at 37 °C with 5% CO_2_ and 95% humidity. Lysis was performed by washing the cells with PBS^-/-^ and scraping them down with FISH buffer (50 mM Tris/HCl pH 7.5, 100 mM NaCl, 2 mM MgCl_2_, 10% glycerol, 20 mM β-glyerolphosphate, 1 mM Na_3_VO_4_, 1× protease inhibitor cocktail and 1% IGPAL). Cells were lysed for 10 min on ice and then centrifuged for 10 min at 15,000× *g*. Supernatant was used for affinity pull down measurements.

### 3.4. GST-Pull-Down

The pull-down of endogenously expressed proteins with purified GST-fused proteins was performed using glutathione agarose beads (#745500.10, Macherey-Nagel, Düren, Germany). Beads were coupled to the GST-fused protein for one hour at 4 °C while mixing and centrifuged for 5 min at 500× *g*. Excess protein was removed by three washing steps. Coupled beads were incubated with HEK293 lysate for one hour at 4 °C on a rotor and again washed 3 times. In the final step, beads were mixed with 1× Laemmli buffer and proteins were denatured at 95 °C for 5 min. Samples were evaluated via SDS-PAGE and western blotting using anti-GST (own antibody, mouse) and anti-IQGAP1 (NBP1-06529, Novus, Wiesbaden Nordenstadt, Germany, rabbit) primary antibodies and secondary antibodies: IRDye^®^ 800 CW anti-Rabbit IgG and IRDye^®^ 680 RD anti-Mouse IgG from LiCor. Values were analyzed by using multiple t test analysis in GraphPad Prism 6 (one unpaired t test per row, fewer assumptions by analyzing each row individually).

### 3.5. Fluorescence Stopped-Flow Spectrometry

All kinetic parameters (k_obs_) evaluated in this study were analyzed using a previously described kinetic analysis protocol [72]. The kinetic parameters were monitored with a stopped-flow apparatus (HiTech Scientific, Applied Photophysics SX20, Leatherhead, UK). The excitation was set for mant at the wavelength of 362 nm, and emission was detected through a cutoff filter of 408 nm. The observed rate constants were calculated by fitting the data as single exponential decay using GraFit program.

### 3.6. Fluorescence Polarization

To determine the dissociation constant K_d_ of direct protein–protein interaction (including weak interactions) fluorescence polarization analysis was performed in a Fluoromax 4 fluorimeter (Horiba Scientific, Loos, France). Here, 1 µM mant-GDP or mant-GppNHp labelled CDC42 proteins were prepared in a total volume of 170 µL in a three directional cuvette. Measurement was performed in polarization mode versus time with an excitation wavelength of 360 nm (slit width: 8 µm) and an emission wavelength of 450 nm (slit width: 10 µm). K_d_ values were calculated in GraFit 5 by fitting the concentration-dependent binding curve using a quadratic ligand binding equation.

### 3.7. GTP Hydrolysis Measurements

GTP hydrolysis rates of a set of different GTPases in presence and absence of GRD1 and its T1046R mutant containing the arginine residue were measured by high-performance liquid chromatography (HPLC) analysis. GTP-bound HRAS in presence of p120 GAP was used as control. Then, 10 μM of each GTPase in the GTP bound state was injected into the HPLC mixing chamber after 1 min of incubation in absence (intrinsic) and presence (GAP stimulated) of 100 μM of GRD1 WT and T1046R variant. The GTP content for each measurement was calculated by dividing the intensity of the GTP detection peak to the sum of the intensities of the GTP plus GDP peaks.

## 4. Conclusions

The exact binding site of the IQGAP GRD and CDC42 is still not completely clear to date. This article provides evidence that the IQGAP GRD does not act as the primary or leading effector binding domain of CDC42 and counterevidence the role of IQGAP GRD in CDC42 binding deduced from a crystal structure of an IQGAP2 GRD2-CDC42Q61L GTP complex. We could show that the GRD does not bind to CDC42 in a nucleotide-dependent manner and that even multiple mutations of the suggested main residues of interaction do not abolish the direct physical interaction in cells and under cell-free conditions. Our data support the binding model of Ozdemir et al. [54] and propose the CDC42 IH as a key binding site for GRD. Furthermore, we shed light once more into the interaction difference of CDC42^wt^ and CDC42^Q61L^ that might be one of the main reasons of the discrepancies in the published data as discussed above. By our comparative measurements of IQGAP1 and IQGAP2 variants, we found differences in their binding strength and specificity towards CDC42^wt^ but also towards various CDC42 variants. Our efforts to investigate also IQGAP3 were so far not successful. The exact binding residues and interaction sites of IQGAP1 and IQGAP2 with the switch regions of CDC42 will still remain to be identified in the future.

## Figures and Tables

**Figure 1 ijms-23-08842-f001:**
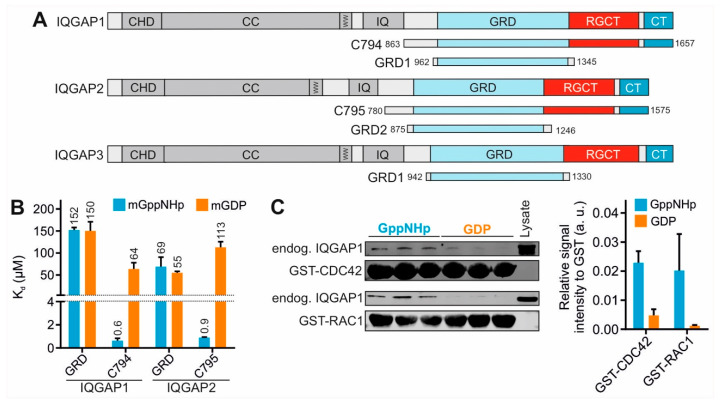
IQGAP GRD binding is nucleotide independent. (**A**) Domain organization of the IQGAP paralogs along with their GRDs and C-terminal fragments assessed in this study (see text for more details). (**B**) Fluorescence polarization analysis of IQGAP1 and IQGAP2 proteins with mGppNHp- and mGDP-bound CDC42. (**C**) Pull-down of endogenous IQGAP1 FL from HEK293 lysates with GppNHp- or GDP-bound GST-CDC42 and GST-RAC1, respectively. Densitometry evaluation of relative IQGAP1 binding to GST-CDC42 proteins (a. u., arbitrary unit) from a triplicate experiment is shown as bar charts.

**Figure 2 ijms-23-08842-f002:**
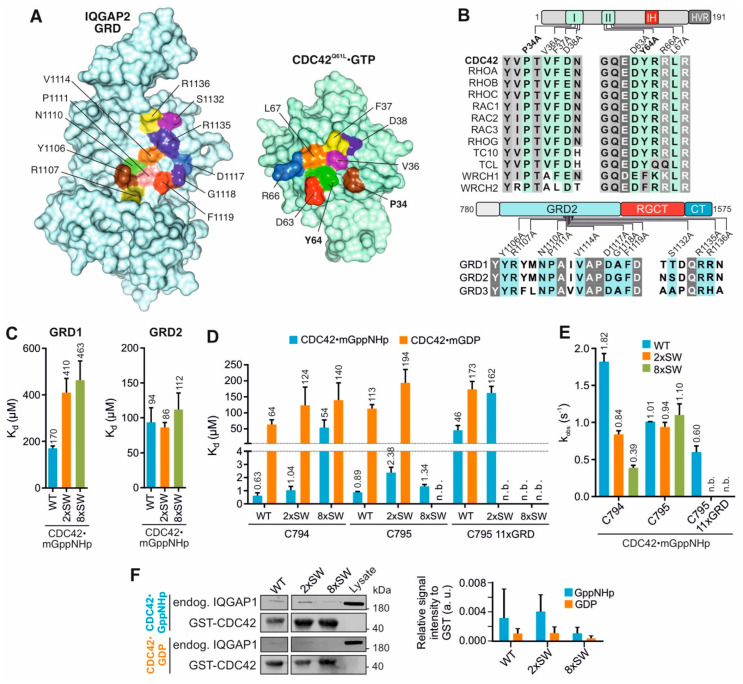
Analysis of CDC42 switch region and IQGAP1 GRD mutants. (**A**) The selection of GRD2 and CDC42 mutations is based on the GRD2/CDC42^Q61L^ structure (PDB: 5CJP). Interacting residues colored on both proteins were selected for mutational analysis. For more details see also Appendix A. (**B**) Multiple sequence alignments of switch regions of RHO GTPases and IQGAPs highlight identical or homologous interacting residues that have been replaced in this study by alanine for analyzing their impact on IQGAP binding. Conserved residues are shaded in grey. Mutations in CDC42 switch regions include 2xSW (bolded residues) and 8xSW (all eight residues, as indicated), and 11xGRD in IQGAP2 C795. (**C**) Fluorescence polarization measurements of mGppNHp-bound CDC42 WT, 2xSW and 8xSW with IQGAP1 GRD1 or IQGAP2 GRD2. (**D**) The K_d_ values for the interactions of IQGAP1 C794, IQGAP2 C795 and C795 11xGRD with the CDC42 variants in mGppNHp- and mGDP-bound form were determined using fluorescence polarization. n.b. stands for no binding observed. C794 and C795 CDC42 WT measurements are included from Figure 1B for simple comparison. (**E**) Observed rate constants (k_obs_) for the IQGAPs association with mGppNHp-bound CDC42 WT, 2xSW and 8xSW were measured using stopped-flow fluorimetry. (**F**) Pull-down of endogenous IQGAP1 FL from HEK293 lysates with GST-CDC42 in GppNHp-bound or GDP-bound state. Cell lysate was used as an input control. Densitometry evaluation of relative IQGAP1 binding to GST-CDC42 proteins (a. u., arbitrary unit) from triplicate experiments is shown as bar charts.

**Figure 3 ijms-23-08842-f003:**
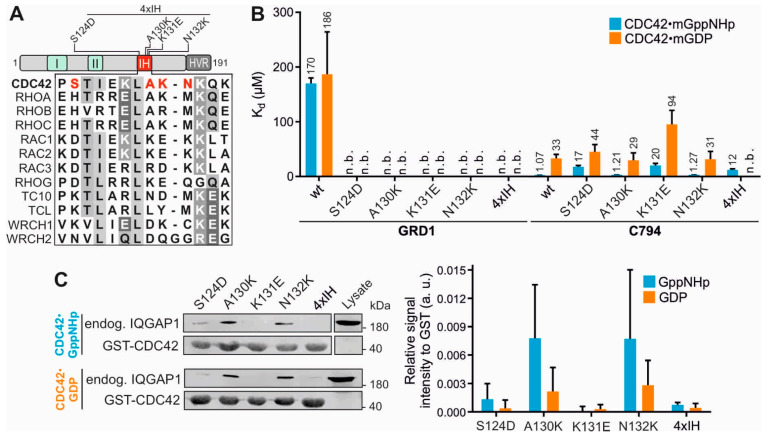
CDC42 IH mutations decrease binding affinity. (**A**) Amino acid alignment of the insert helix of selected members of the RHO GTPase family. CDC42 mutations (red) to RAC1 were introduced outside of the conserved regions (grey). (**B**) Fluorescence polarization data for the interaction of GRD1 and C794 with the CDC42 IH variants. (**C**) Pull-down of endogenous IQGAP1 FL from HEK293 lysates with GST-CDC42 IH variants in both GppNHp-bound and GDP-bound forms. Cell lysate was used as an input control. The pull-down data for GST-CDC42^wt^ is shown in Figure 2F as all pull-down experiments were conducted under the same conditions. Densitometry evaluation of relative IQGAP1 binding to GST-CDC42 proteins (a. u., arbitrary unit) from a triplicate experiment is shown as bar charts.

**Figure 4 ijms-23-08842-f004:**
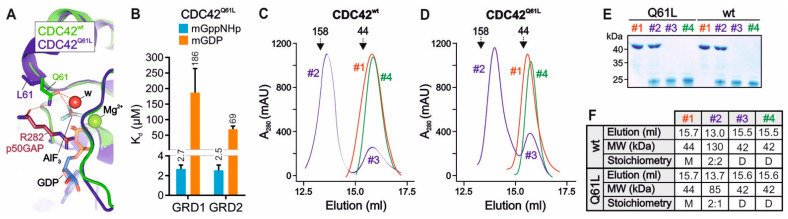
Comparative analysis of IQGAP1 GRD1 interaction with CDC42^Q61L^ and CDC42^wt^. (**A**) Structural overlay of CDC42^wt^ GDP AlF_3_ p50GAP (green; PDB: 1GRN) on CDC42^Q61L^ GTP IQGAP2 (blue; PDB: 5CJP) with the focus on Q61 hydrogen bonds (red dashed lines). GDP AlF_3_ mimics the transition state of the GTP hydrolysis reaction and is coordinated with the magnesium ion (Mg^2+^) and the nucleophilic water molecule (w) and the arginine finger (R282) of p50GAP. Aluminum trifluoride (AlF_3_) mimics the γ-phosphate of GTP in the transition state. In contrast to L61, Q61 is critical for the catalysis of the GTP hydrolysis reaction through three hydrogen bonds (see text). (**B**) Fluorescence polarization data of IQGAP GRD1 with CDC42 mGppNHp and CDC42 mGDP. (**C**–**F**) IQGAP GRD differently forms complexes with CDC42^WT^ and CDC42^Q61L^, respectively, when applied on an analytical SEC. For this purpose, CDC42^WT^ GppNHp (**C**) or CDC42^Q61L^ GppNHp (**D**) were mixed with IQGAP1 GRD1 and SEC was performed on a Superdex 200 10/300 column using an ÄKTA purifier (flow rate of 0.5 mL/min, fraction volume of 0.5 mL) and a buffer, containing 30 mM Tris/HCl, pH 7.5, 150 mM NaCl, and 5 mM MgCl_2_. The elution profiles represented one peak for the respective CDC42 proteins (#1), two peaks for the respective mixtures of respective CDC42 proteins with GRD (#2 and #3) and one peak for the GRD1 (#4). (**E**) Coomassie brilliant blue staining of the corresponding elution volumes indicated that only peaks #2 contain GRD1 complexes with CDC42^WT^ or CDC42^Q61L^, respectively. Peaks #3 only contain the CDC42 proteins as compared to the peaks #1 and #4. (**F**) The SEC profiles of CDC42^WT^ and CDC42^Q61L^ are summarized for each peak regarding the elution volume, the molecular weight (MW) and the stoichiometry. M stands for monomeric and D for dimeric. The theoretical MWs of CDC42 (21.2 kDa) and GRD (43 kDa) were calculated using the Expasy Protparam tool. The presented MWs for each peak was calculated based on the calibration curve (aldolase 158 kDa and ovalbumin 44 kDa, respectively) and partition coefficient plot (Kav = Ve − V0/Vc − V0) versus the logarithm of MWs; Ve: elution volume number; V0: void volume (8 mL); Vc: geometric column volume (24 mL)). Accordingly, peaks #2 correspond to a heterotetrameric complex between CDC42^WT^ GppNHp and GRD1 with a MW of 130 kDa, and a heterotrimeric complex of GRD and CDC42^Q61L^ GppNHp with a MW of 85 kDa.

**Figure 5 ijms-23-08842-f005:**
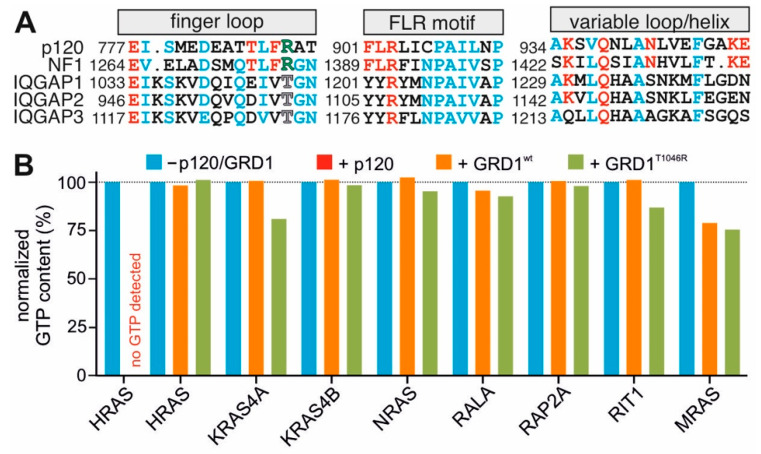
Deviation in RAS-binding residues in GRDs cause lack of RASGAP activity. (**A**) Sequence alignment of human RASGAPs p120, neurofibromin (NF1) and the three IQGAP paralogs highlights distinctive deviations in three signature motifs (grey boxes): the finger loop, FLR region and a7/variable loop. RAS-binding residues are shown in red and conserved residues in blue. The catalytic arginine (green) is substituted by threonine in IQGAPs. The numbers correspond to the amino acids of the respective proteins. (**B**) GTP hydrolysis of various RAS family GTPases was measured in the absence (blue) and in the presence of p120 GAP domain (red; positive control, where no GTP detected) or GRD1^wt^ (orange) and GRD1^T1046R^ (green). The GTP hydrolysis of the RAS proteins (10 µM) was measured via HPLC and the GTP content normalized to 100% before adding p120 or GRD1, respectively, at 100 µM concentrations and 1 min incubation time.

## Data Availability

All the data are in the manuscript.

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
