# Peer review of "CDC42-IQGAP Interactions Scrutinized: New Insights into the Binding Properties of the GAP-Related Domain"

_ijms, 2022, doi:10.3390/ijms23168842_

Round 1

Reviewer 1 Report

The manuscript by Mosaddeghzadeh et al. uses recombinant proteins to analyze the interaction of the three human IQGAP proteins with the small GTPase Cdc42. In a profound biochemical approach, the authors compare the binding of different domains within the C-terminus of IQGAP1, -2 or -3 for binding to guanosine di- or triphosphate-loaded Cdc42. In a comparative analysis, they determine the dissociation constants for either the GRD domain alone or for the GRD-RGCT-CT domain assembly. The authors conclude that previous studies on the structural basis of Cdc42-induced dimerization of IQGAPs may not fully reflect these multiple interaction sites of IQGAPs with the GTPase, while the contribution of the insert helix (IH) specific for Rho GTPases and the contribution of the RGCT domain within IQGAPs to the binding interaction may have been overlooked.

To assess the results obtained in the binding analyses, I consider chapter 2.4 to be of particular importance. The authors show here one other time that the hydrolysis impaired Q61L mutant is not a suitable analogue for triphosphate bound Cdc42, as the switch II interface is largely impacted by the mutation. As a results, the Q61L mutant shows a 30-50-fold reduced binding affinity to an effector protein compared to the wild-type GppNHp-bound state, although the Q61L mutant protein is likewise in the GTP-bound state. The use of the Q61L mutant as a “cheap” means to generate a triphosphate loaded GTPase might explain discrepancies in previous findings.

The study appears to me as an important contribution to the interaction of IQGAPs with the GTPase Cdc42, that particularly highlights the contribution of the insert helix in Cdc42 to the specificity of the interaction. I have only some minor points that demand revision.

Criticism:

It seems not appropriate to this reviewer, that the authors show the panels with dissociation constants (Kd) with a linear scale on the Y-axis. I strongly recommend using the log10 scale to show the differences in binding affinity.  This holds for Fig. 1B, 2C, 2D, 3B, and 4B.  Linear scale is not appropriate.

Line 15:  … evolutionarily conserved …

Line 177:  Sentence?  … proposed that Cdc42-GTP-GRD is a GTPase-effector complex.

Line 385:  What is the a7/variable loop?  A7 is not introduced.

Author Response

Thank you very much for your time and efforts in reviewing of our manuscript, and for your positive assessment. We have revised our manuscript point-by-point according to your suggestions as followed. Changes in the manuscript are highlighted in yellow.

Reviewer 2 Report

The authors of this paper have tried to solve a discrepancy in literature regarding GRD dependent or independent binding of CDC42 and IQGAP. The authors here through extensive mutational analysis as well as full length protein binding studies, for binding between the CDC42 and IQGAP try to answer this question. The authors provide enough evidence to showcase the need of GRD as a scaffold for binding of IQGAP to CDC42. Similarly they verify that the effector domian for IQGAP for CDC42 being RGCT and not GRD.

The authors also clearly point out the reason for the discrepancy though an excellent understanding of mutational substitution analysis. Overall the manuscript provides a clearer picture towards understanding the mechanistic activity of binding between IQGAP and CDC42.

Here are some of the changes expected from the authors in the manuscript:

1. A major oversight by the authors was: both the method of statistical analysis and statistical significance for all figures to be missing and must be reported by the authors

2. Although the authors showcase binding using pull-down assays and fluorescence polarization. A more direct way for determining binding would be do a SPR experiment between the purified proteins, especially for some of the critical binding experiments.

3. Catalog numbers of materials and reagents are missing.

Author Response

(The authors gave the same response as above.)
